# Off-chip prefetching based on Hidden Markov Model for non-volatile memory architectures

**Adrián Lamela[1]☯, Óscar G. Ossorio[2], Guillermo Vinuesa[2], Benjamín Sahelices[1]☯***

**1** Department of Computer Science, School of Informatics Engineering, University of Valladolid, Valladolid, Spain, **2** Department of Electronics, School of Informatics Engineering, University of Valladolid, Valladolid, Spain

☯ These authors contributed equally to this work.
* benja@infor.uva.es

**Data Availability Statement:** All data can be freely accessed via this public github repository: https://github.com/adrianlamela/HMM_Results.

**Funding:** The work was supported by the Spanish Ministry of Economy and Competitiveness (Grant

## Abstract

Non-volatile memory technology is now available in commodity hardware. This technology can be used as a backup memory for an external DRAM cache memory without needing to modify the software. However, the higher read and write latencies of non-volatile memory may exacerbate the memory wall problem. In this work we present a novel off-chip prefetch technique based on a Hidden Markov Model that specifically deals with the latency problem caused by complexity of off-chip memory access patterns. Firstly, we present a thorough analysis of off-chip memory access patterns to identify its complexity in multicore processors. Based on this study, we propose a prefetching module located in the LLC which uses two small tables, and where the computational complexity of which is linear with the number of computing threads. Our Markov-based technique is able to keep track and make clustering of several simultaneous groups of memory accesses coming from multiple simultaneous threads in a multicore processor. It can quickly identify complex address groups and trigger prefetch with very high accuracy. Our simulations show an improvement of up to 76% in the hit ratio of an off-chip DRAM cache for multicore architecture over the conventional prefetch technique (G/DC). Also, the overhead of prefetch requests (failed prefetches) is reduced by 48% in single core simulations and by 83% in multicore simulations.

## Introduction

Non-Volatile Memory architectures have very high memory density with very low energy consumption, which allows for massive data sets in main RAM memory that can be directly accessed by the cores [1–6]. Nevertheless, its main problem is that access latency increases by a factor of 2x to 4x, worsening the memory wall problem [6–9]. Storage-class non volatile main memory enables RAM with higher density and lower energy so it can outclass traditional memory architectures [5, 6] if its latency problem is solved. One way to reduce latency is to use a hardware-managed external DRAM cache located between on-chip caches (L1-L2-LLC) and main non-volatile memory [6, 10].

No. TEC2017-84321-C4-2-R) with support from Feder Funds and also by MINECO/AEI/ERDF (EU) (grant PID2019-105660RB-C21 / AEI / 10.13039/ 501100011033), Aragón Government (T58_20R research group), and ERDF 2014-2020 "Construyendo Europa desde Aragón". The funders had no role in study design, data collection and analysis, decision to publish, or preparation of the manuscript.

**Competing interests:** The authors have declared that no competing interests exist.

Off-chip caches based on SDRAM technologies improve latency and bandwidth, but the filtering effect from on-chip cache hierarchy decreases its efficiency because it hides spatial and temporal locality. Poor locality reduces DRAM cache hit ratio [11] and low hit ratio decreases bandwidth and increases cache latency, thus, different and more complex prefetch techniques are needed in order to manage off-chips caches.

Hardware prefetch is a very well-known technique which hides latency by increasing hit ratio [12–16]. This technique keeps track of the accessed locations to make a prediction of its future usage so it can bring them to the cache in advance and timely. Hardware prefetch is very common in the on-chip cache hierarchy of commercial processors [17–19]. Off-chip DRAM cache has some specific characteristics which differ from on-chip caches, like its gigascale cache capacity and very high bandwidth that makes them very good candidates to apply complex hardware prefetch techniques along with eager data movement with the aim of increasing bandwidth and reducing latency.

Off-chip caches receive very reduced and filtered information of the programs behavior because much of the temporal locality is filtered on the on-chip caches, and (depending on its size and geometry) some of the spatial and algorithmic locality is also filtered. Moreover, as off-chip caches are shared with all cores, the sequence of misses and replacements received correspond to different programs running simultaneously. This complicates even more the task of keeping track the lines used by programs. For all these reasons, it is worth to develop and test more complex prefetch techniques which could model memory references via Markov transition diagrams [20] to predict the most probable future address sequence based on the past references. A lot of similar systems have been proposed since Markov Predictors [21], some of them with improved data structures that save memory and increase efficiency [14–16, 22, 23].

Several non-volatile memory (NVM) architectures are currently under intense research because they have good characteristics to become the main RAM storage in future systems. They are expected to have terabytes of capacity, can directly host file systems like Linux DAX and its zero power consumption when idle makes them ideal to use in big data centers [16]. One of the most promising architectures (FAM [16]) decouples NVM modules from compute units to improve utilization, sharing and bandwidth but also highlights the relevance of the latency problem. Several approaches [14, 15] use hardware prefetching to deal with the problem of high latency in LLC misses. Overall, it is clear that off-chip latency is a very relevant problem that must be solved to design efficient memory hierarchies.

In this work we discuss the case of hardware prefetching for off-chip DRAM cache to minimize latency when running multiple simultaneous applications with very complex off-chip data access patterns. We also make an statistic analysis of off-chip access patterns that further motivates the need of a prefetch tool capable to deal with complex memory access patterns. We develop and evaluate a proposal of hardware prefetcher based on a Hidden Markov Model. Specifically, our work makes the following contributions:

- We analyze the off-chip memory access profile and extract and study its characteristics.

- We develop an statistical model based on a Hidden Markov Model to keep track of groups of off-chip accesses triggered by the different threads running on the multicore processors.

- We propose an specific solution for Hidden Markov Model sequence estimation with $O(n)$ complexity.

- We quantify the hit ratio improvement for off-chip caches and the accuracy of the proposed technique.

The paper is organized as follows: Section *Off-chip Memory Accesses* provides an analysis of main memory accesses patterns and characteristics that motivates our work; Section *Related Work* discusses similar state-of-the-art proposals; Section *Prefetch Based on Hidden Markov Model* details the proposed prefetch tecnique and describes a hardware implementation; Section *Evaluation* provides the evaluation environment and results of our models; and Section *Conclusions* presents the outcome of this work.

## Off-chip memory accesses

On-chip cache hierarchy in modern processors is very efficient for filtering most of the temporal locality and, when a good prefetch algorithm is used, some of the spatial and algorithmic locality. This efficiency means that most of the locality is filtered in the on-chip memory hierarchy. As a result, off-chip caches receive fragmentary locality information and its efficiency is reduced. Off-chip caches are necessary in modern memory hierarchies due to the use of high density and long latency backup memories, such as non volatile RAM. Previous studies show that an specific improvement must be done on these caches to avoid a negative impact in memory hierarchy [11, 24–26].

In this section we analyze off-chip memory accesses to unveil their specific characteristics. Off-chip memory accesses are modelled using data collected from real executions of 20 applications, 17 SPEC CPU2006 and 3 specific applications. The data for this section has been obtained using the methodology described in section *Evaluation*. Our analysis has been performed in three dimensions, temporal locality, spatial locality and algorithmic locality. Results from this study are used to justify the proposal of more complex DRAM cache management strategies in order to improve both hit ratio and latency.

### Temporal locality (TL)

In this section we show that LLC filters most of the temporal locality and, as a consequence, the relevant off-chip memory access patterns become more complicated. The different sources of off-chip accesses are: regular (data/instructions) memory accesses that misses on LLC, software/hardware core-side prefetches, and replacements triggered by the first two types. So there are two different sources of LLC misses and a third one indirectly related to them and all three types depend upon the data access patterns of the running applications and the size, geometry and policy of a hierarchy of cache memories. Depending on the efficiency of this on-chip cache hierarchy, locality information carried by LLC misses can be very limited and fragmentary. To prove this assumption we show temporal histograms representing time interval probability between consecutive off-chip accesses to the same location. We also provide the number of such accesses. See Fig 1(a)–1(c) for benchmarks *lbm*, *libquantum* and *omnetpp*. For each figure, four LLC configurations are used with different size and associativity (size 16MB and 32MB and associativity 1 and 8). The repetition interval is represented as a percentage of the *forget threshold*, which is the limit of time in which a repetitive access to the same line is considered as temporal locality. In those figures 100 represents the *forget threshold*.

Fig 1(a) shows that *lbm* benchmark is much more sensitive to associativity than to capacity, so its main miss category is conflict-type. With 32MB size and 8 associativity, LLC is very efficient in *lbm* benchmark, filtering 99% of misses. One interesting result of this filtering is that the main locality behavior in 16MB and Assoc-1 LLC cache is lost and instead a lot of complex and temporarily dispersed repetitive patterns appear, as seen in the 32MB and Assoc-8 configuration. Fig 1(b) shows *libquantum* benchmark. This application is very sensitive to cache capacity but does not change when associativity is increased. As it can be seen in the figure, LLC filters 94% of misses, and also, with the best LLC configuration, a very complex pattern is

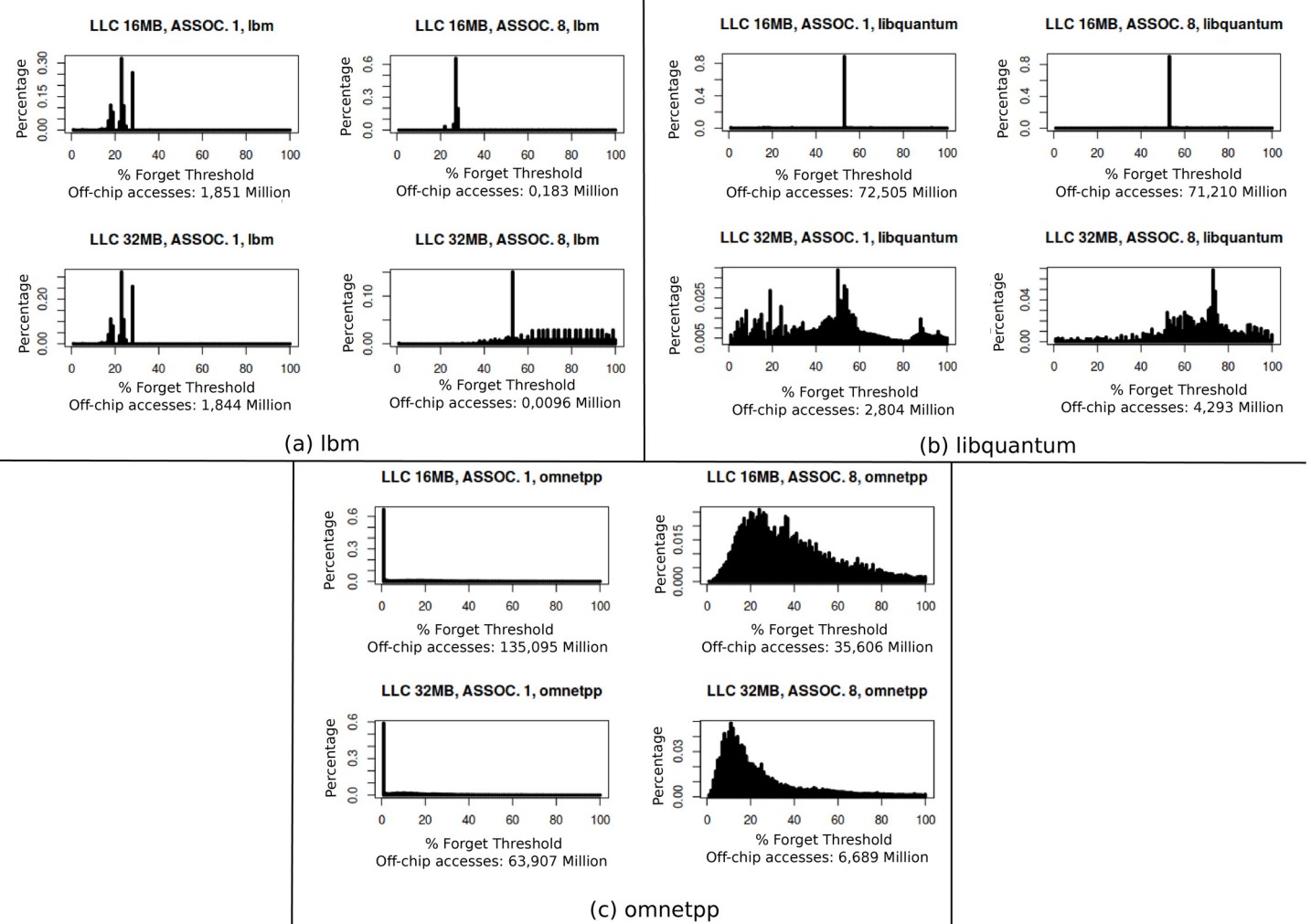

**Fig 1. TL with four LLC configurations.** Temporal locality of *lbm*, *libquantum* and *omnetpp*, represented as time interval probability between consecutive accesses (100 in this figure is the forget threshold), with four different LLC configurations: Size 16MB-32MB and associativity 1-8. The number of off-chip accesses is included showing the ability of LLC to filter cache misses.

observed. The same behavior is observed in Fig 1(c) for *omnetpp* benchmark. In this case, the benchmark is sensitive to both LLC size and associativity, but the same complex pattern appears with the best LLC configuration.

These three applications are representative of different types of misses, but all three have in common a deep reduction in the number of repetitive accesses to the same locations in off-chip accesses and a clear increase in complexity of access patterns. This behavior prevents DRAM cache to have good hit ratio and latency. When the main temporal locality behavior is removed in the LLC, the remaining memory accesses show a significant rise in relevance, as shown in Fig 1. Because of their complexity these accesses are not filtered in the LLC, thus they require more sofisticated off-chip cache management heuristics.

## Spatial locality

Processes usually have very complex memory access patterns: they use several memory areas at the same time and the number and location of those areas change through time. Most of these

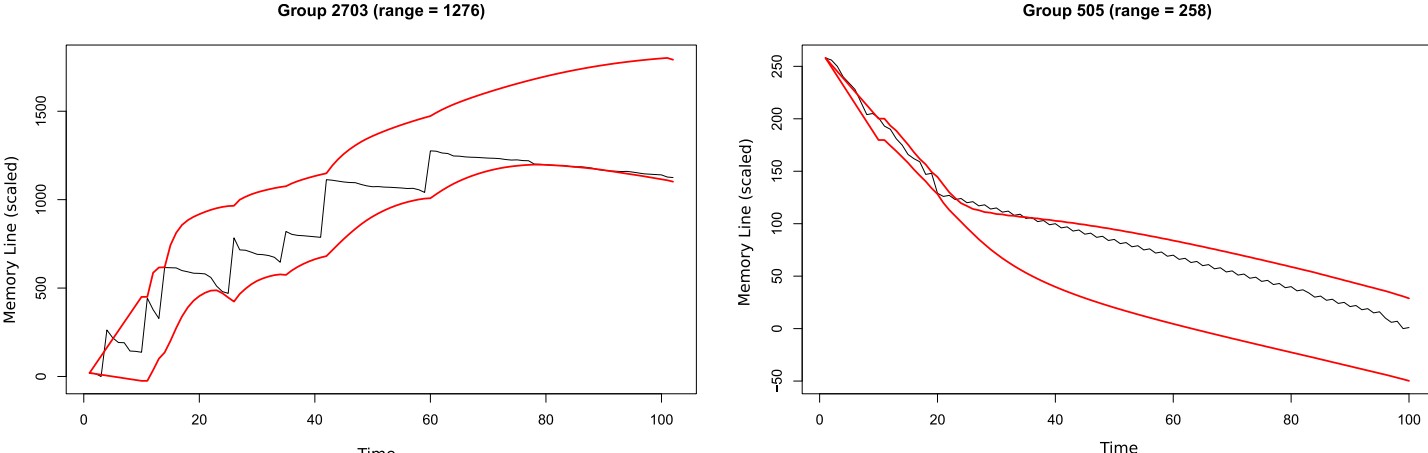

**Fig 2. Example of two groups identified by** HMM **in mcf benchmark.** The figures represent the off-chip accessed lines through time and the address interval (red lines) that our prefetcher identify and may use to trigger DRAM cache prefetches. These groups appear simultaneously and are identified, isolated and grouped by our HMM proposal so prefetches may be individualized to each group.

complex memory patterns come from linked data, indirect indexing and hash functions. These types of data structures reduce spatial locality and, consequently, cache performance. We use the *mcf* benchmark as an example of such behavior (see results on section *Evaluation*). Fig 2 shows two examples of simultaneous off-chip access groups identified and isolated by the prefetch technique proposed in this work, HMM. As it can be seen, spatial locality exists, but it is complex since it replicates an algorithmic behavior. For each application, several groups like the shown in Fig 2 can appear at the same time, and in a multicore this is multiplied by the number of cores. For this reason, we analyze this behavior as algorithmic locality in the next section.

## Algorithmic locality

Algorithmic locality defines the memory access patterns related to the use of complex data structures and flow control of the applications. Sometimes this behavior is very complex, and therefore it is very difficult for prefetchers to identify it. We use frequency analysis of off-chip memory accesses as a model of representation. Under this analysis, the frequency of main memory references is discretized and represented so the program behavior related to memory access patterns is modelled through its execution time. To represent this behavior we use *off-chip operations per kilo-cycle (* OPKC) as the basic metric. The discretized representation of OPKC through time reflects the behavior of the corresponding code.

Fig 3(a) shows the evolution of off-chip OPKC for *mcf* benchmark through its running time with the four LLC configurations. As it can be seen, in the last third of the represented interval, data access pattern changes very quickly and LLC cannot deal with the complexity in any of its configurations, so there is a burst of LLC misses, represented by OPKC. Fig 3(b) shows hit ratio for a proposal of DRAM cache (see results on section) through the same time interval, comparing our prefetcher HMM vs. a common G/DC prefetcher [22]. When algorithmic complexity begins to grow starting on cycle 1100-Million, HMM deals correctly with complexity and keeps high hit ratio while G/DC does not. The reason for this good performance is that HMM is able to identify complex simultaneous groups, as shown in Fig 2.

In Fig 4 we show off-chip OPKC evolution for *lbm*, *libquantum*, *omnetpp* and *milc* with four LLC configurations. *lbm* benchmark, represented in Fig 4(a), has such a large working set size

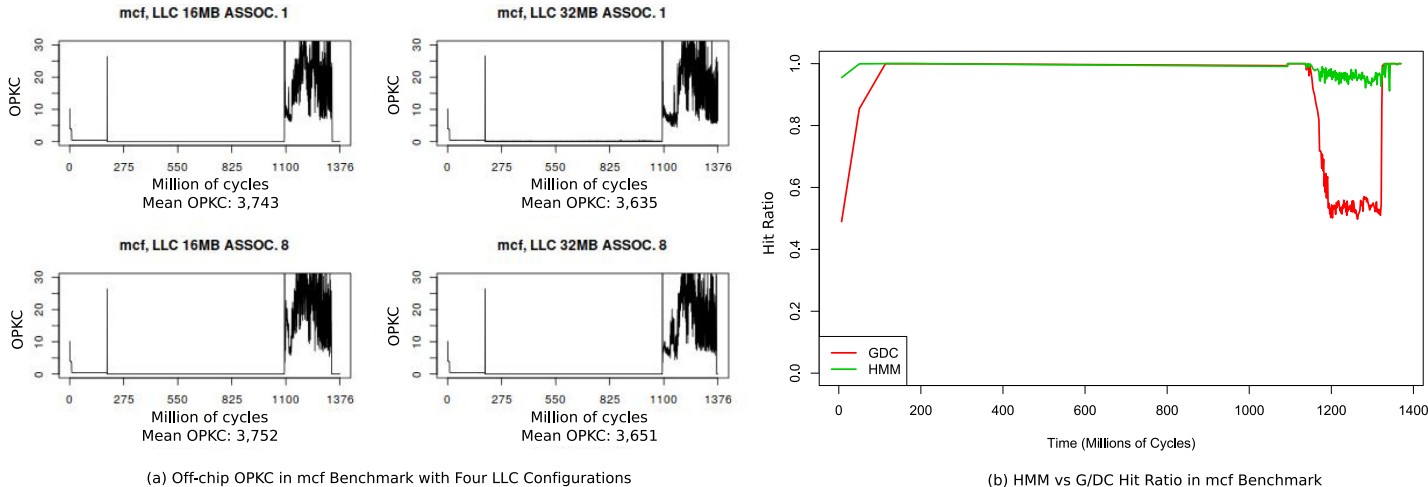

(a) Off-chip OPKC in mcf Benchmark with Four LLC Configurations

(b) HMM vs G/DC Hit Ratio in mcf Benchmark

**Fig 3. Analysis of spatial locality in *mcf* benchmark.** Spatial locality is modeled using off-chip OPKC, so when OPKC increases hit ratio in external cache becames critical. In (a) can be seen how starting on cycle 1100-Million OPKC increases in all LLC configurations. (b) shows how starting on cycle 1100-Million our proposal HMM achieves very good hit ratio when off-chip cache presure increases, beating clearly G/DC prefetch technique in this scenario.

**Fig 4. Frequency analysis with four LLC configurations.** In this figure we use off-chip OPKC to describe the types of misses related to LLC size and associativity. *lbm* OPKC is independent of LLC configurations, *libquantum* misses reduce with LLC size so they are capacity type, *omnetpp* misses diminish when associativiy increases so they are conflict type and, finally, *milc* has both types.

that LLC does not filter most of their accesses. Average OPKC are roughly the same for the four different LLC configurations. Otherwise *libquantum*, *omnetpp* and *milc*, see Fig 4(b)–4(d), show high sensitivity to LLC size and associativity. In *libquantum* benchmark, capacity misses are dominant, most of *omnetpp* misses are conflict type and *milc* has both, capacity and conflict. For all benchmarks but *lbm*, OPKC reduces broadly with better LLC configurations and so the requests reaching DRAM cache are also reduced. This leads to a strong need of developing complex prefetch techniques for DRAM cache to deal with the complex off-chip access patterns.

## Related work

The identification of spatio-temporal locality is the heuristic that the majority of the prefetchers use to increase performance of cache memories. Different methods have been proposed in order to identify it. Joseph and Grunwald [20] proposed a Markov prefetcher that allows the identification and prefetch of multiple sequences of memory misses. This idea requires simple hardware while allowing for very good miss reduction rate. Several proposals based on Markov appear in the bibliography.

Global History Buffer (GHB) is a simple data structure in which Nesbit and Smith [22] base a very effective prefetch technique. This proposal allows to correlate the sequence of miss addresses using different location methods (as PC and address) and three detection mechanims (constant stride, delta correlation and address correlation). This is an effective and simple solution, where the performance of which is comparable with other similar techniques while needing simpler hardware support.

Somogyi et al. [27] proposed Spatial Memory Streaming (SMS), which allows for identifying access patterns that are not always consecutive. They use contextual data as PC to do it. This technique was proved successful when identifying complex access patterns. Peled et al. [12] developed Semantic Locality and context-based prefetching as a generalization of SMS. It uses hardware context information plus compiler injected hints to feed a machine learning technique which predicts future memory accesses of irregular data and algorithms. This is a complex technique that has good results with both simple and complex locality patterns.

eDRAM LLC prefetch is a similar approach which focus on irregular granularities to predict irregular memory streams from large working sets in memory-intensive applications [14]. It contains a prefetch buffer (history table) between LLC and main memory and a controller that implements three different prefetch modules to find address deltas among irregular memory streams. The history table stores overall information of memory accesses divided in a page table, line table and picture table, to detect different access sizes. This technique has good results dealing with size complexity but does not adapt well to complex address patterns.

Prodigy [15] uses hardware-software codesign to deal with the complexity of data-indirect irregular workloads. It uses static information from programs and dynamic information from hardware to extract program semantic data which are used to generate prefetch requests. Prodigy gets very good results in energy and latency compared with standard prefetch techniques. Our proposal succeeds in capturing irregular data access patterns without the use of compiler information from running programs.

Pre-FAM [16] is targeted to fabric-attached non-volatile memory architectures. This kind of organization is very sensitive to latency since NVM modules are far from processing modules. This work shows the relevance of prefetching LLC misses in NVM architectures. It uses an address map pattern prefetcher dividing memory in multiple zones keeping one entry on a table for each one. This approach limits the prefetch to the identified zones being less adaptative than our proposal, which dynamically identifies the active zones and adapts to dynamic changes.

In summary, some of the proposals show good results identifying complex spatio-temporal and algorithmic locality but they require a lot of context information and complex algorithms. Other proposals are simpler but fail when there are multiple memory sequences with complex spatio-temporal relations. We propose to use only available information (miss addresses) combined with a complex technique to identify different simultaneous regions and successfully dispatch prefetches on each one.

## Prefetch based on Hidden Markov Model (HMM)

In previous sections we have shown how LLC misses show very complex repetitive patterns. To deal with this complexity we define the term *group* as a set of addresses that share a pattern that may be recognized with linear regression models. An analysis of several simultaneous address groups is carried out for each core. This task must be performed using virtual addresses (VA) instead of physical addresses (PA). The reason is that VA addresses contain information of the working set as they reflect the virtual address spaces currently in use. Instead PA suffers a previous translation phase that allows for an efficient physical memory management but hides some of the information related to the current working set. The memory architecture we propose is shown in Fig 5, in which all the cache hierarchy is based on virtual addresses and, accordingly, the prefetch analysis is performed using virtual addresses instead of physical ones. This allows to get more precise information about different memory groups in the current working set. Besides, when VA are used, the number of virtual to physical translations is greatly reduced because it is done only for LLC misses. Another positive consequence of using virtual memory hierarchy is that the critical translation overhead between the core and the L1

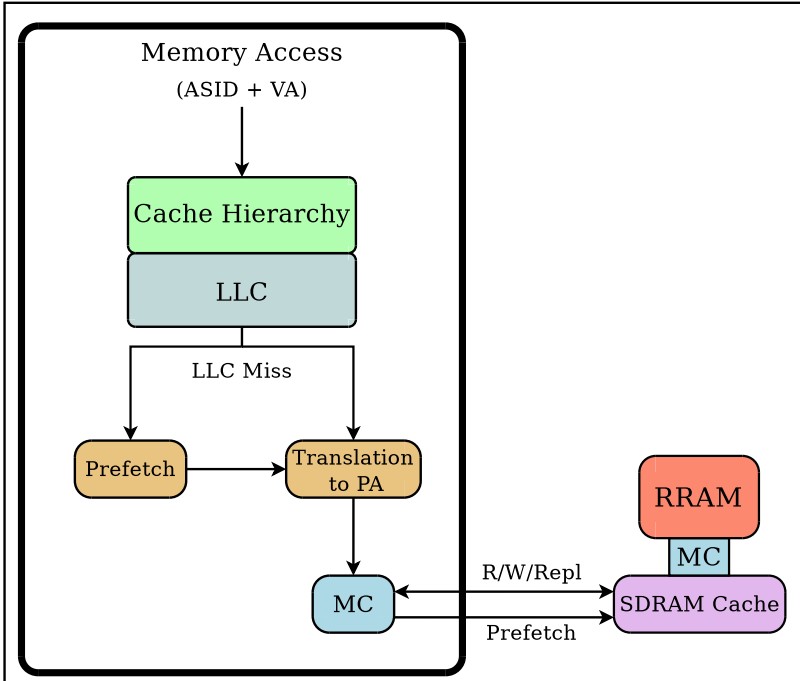

**Fig 5. Schematic architecture.** Proposed virtual address (VA) based architecture for off-chip prefetching. The use of VA allows the prefetcher to exploit all the locality information at the cost of increase memory and energy use to store tags and ASID information. The number of VA to PA translations is greatly reduced due to the positive effect of the cache hierarchy in reducing off-chip accesses. Off-chip prefetchers move data/instructions in advance from NVM-RAM to DRAM cache.

is eliminated, but at the cost of increased tag memory size in cache hierarchy to store ASIDS. This increase in memory overhead implies a growth in energy waste. An example of this kind of virtual memory architecture is shown in [28].

On a certain time interval, each process simultaneously accesses to several different groups in the working set. We use the sequence of LLC misses to identify each group and, once the groups are identified, each miss is assigned to the nearby group. There are several types of proposals in the literature that implement hardware prefetching based on different heuristics. Temporal data prefetchers have data structures that efficiently identify the repetition of stream misses; spatial data prefetchers rely on address correlations to identify future accesses, and spatio-temporal data prefetchers mix both techniques [13, 14, 21, 22, 27, 29–31]. Their main limitation is its lack of ability to identify high complex miss data patterns relationships between multiple simultaneous groups. This is very important in NVM architectures because its high capacity allows for big data and other complex applications. The Hidden Markov Model (HMM) provides the mechanism for grouping LLC misses related with complex address patterns. Based on the assumption that misses can be characterized by groups, the next accessed lines depend directly upon its group. Hence, once the sequence of values depending on the hidden states are known, the hidden sequence can be estimated, as it can be seen Fig 6. This model assumes the change of groups is based on a certain probability and the memory line depends upon the group. This means that once the line is observed we can estimate the hidden state, i.e. the group. We believe this is a very natural way to deal with the hardware prefetch problems in high complexity environments like an off-chip cache in a NVM-RAM memory architecture.

As an example, in Fig 7(a) we show the address line numbers that miss on LLC on a certain time interval for *astar* benchmark. To isolate the different groups of misses, we remove the upper side and represent only the lower values in Fig 7(b) which shows new miss lines that were hidden in the previous figure. In this figure it is easier to identify several different groups of lines. In Fig 7(c) all the identified groups are represented with different colors (color black represents misses not grouped). The identified groups are broken down in Fig 8 in which four different groups are identified and represented. Fig 9, shows the prediction interval of a linear model over time for the identified groups. In the next subsection we explain how HMM performs this group identification.

## Hidden Markov Model for VA clustering

Suppose $Y(t)$, $t = 1, \ldots, T$ are the lines of memory accessed by LLC misses, and $S(t)$, $t = 1, \ldots, T$ is the group of each of the lines. We define Hidden Markov Model as a quadruple (Q,V,A,B) where:

- $Q = \{q_1, \ldots, q_N\}$ is the set of states with unknown $N$.

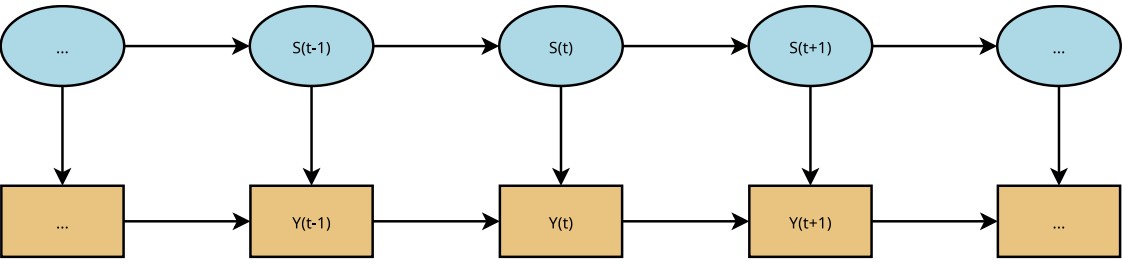

**Fig 6. Symbolic representation of Hidden Markov Model (HMM).**

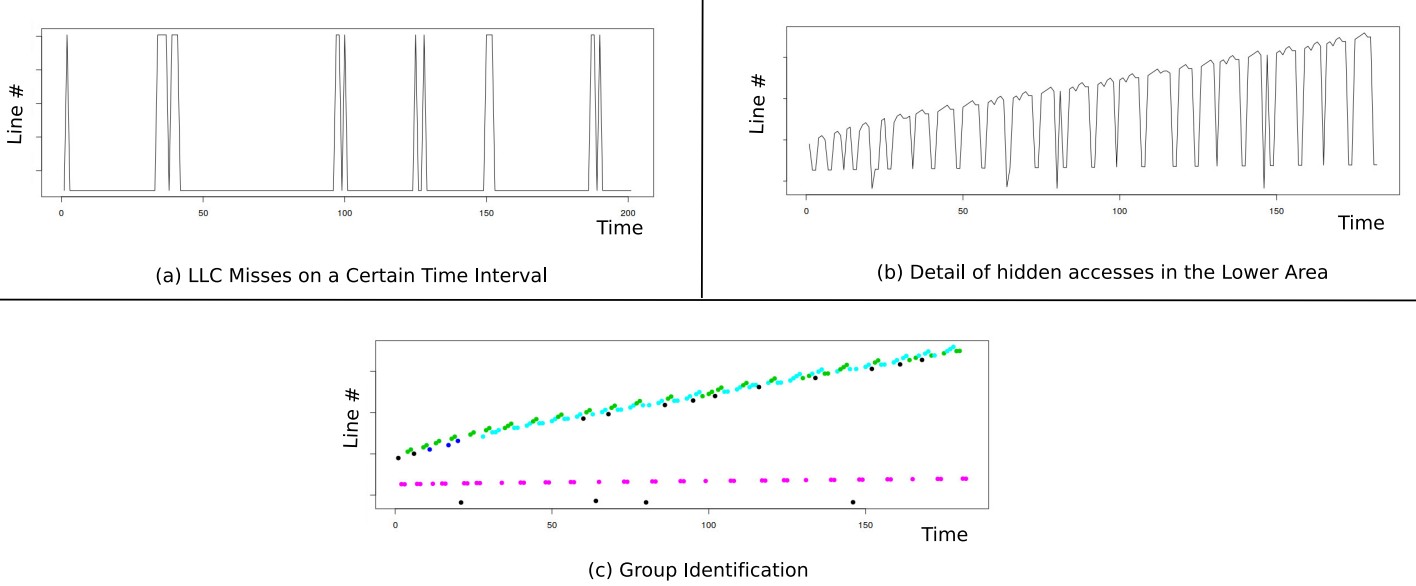

(a) LLC Misses on a Certain Time Interval

(b) Detail of hidden accesses in the Lower Area

(c) Group Identification

**Fig 7. Example of *astar* spatial locality.** Algorithmic complexity in *astar* with multiple simultaneous groups that are isolated and identified by our HMM proposal. In (c) the different groups are represented by colors.

- $V = \{v_1, \ldots, v_M\}$ is the set of observable values, that is, the memory lines. Each line is 64-B long so $V \subseteq \{0x000000000000, \ldots, 0xFFFFFFFFFFC0\}$.

- $A = \{a_{ij}: q_i, q_j \in Q\}$, where $a_{ij} = P(S(t) = q_j | S(t-1) = q_i)$ is the transition probability from $q_i$ to $q_j$. They are unknown but based on Markov's property which states that only the immediate past matters.

- $B = \{b_j(v_k): q_j \in Q, v_k \in V\}$, where $b_j(v_k) = P(Y(t) = v_k | S(t) = q_j)$. We suppose that $Y(t)|S(t) = q_k \sim \mathcal{N}(\mu_k(t), \sigma_k^2(t))$ in order to compute the estimations, as described later in this section.

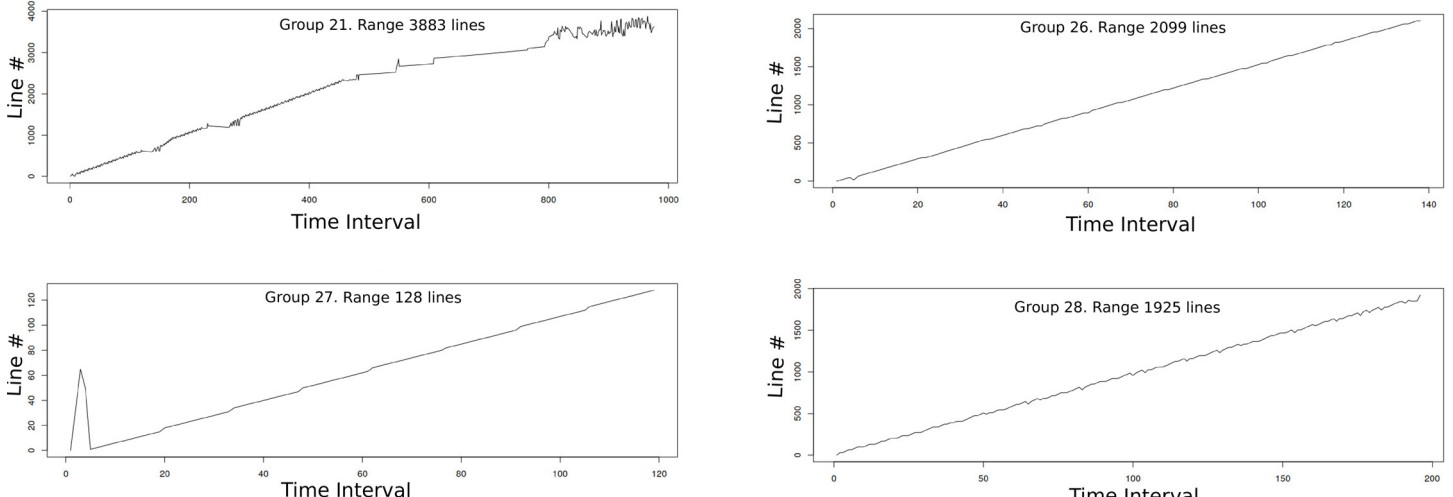

**Fig 8. Main areas of spatial locality in *astar*.** Example of four groups identified in *astar* by HMM. This information is fed to the prefetcher to get intervals of addresses with high probability of future use.

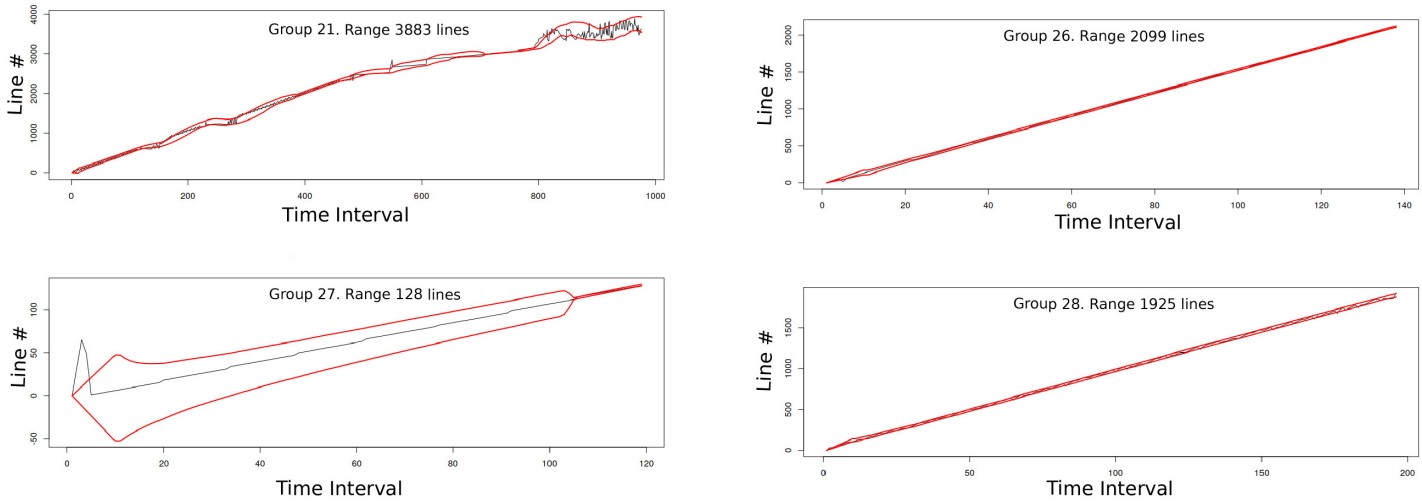

**Fig 9. Main areas of spatial locality in *astar* with recognized intervals.** Based on the group identification, HMM gets address intervals with high probability of use, which are shown in this figure between the red lines.

This is a particular definition of the well known Hidden Markov Model with the only difference that we do not need the initial probabilities because we assign the lines in order to consecutive groups.

The procedure is based on the estimation of the sequence $S(t)$, $t = 1, \ldots, T$ using the information in $Y(t)$, $t = 1, \ldots, T$. Although there are solutions for this problem in the bibliography like the Baum-Welch algorithm [32], one of the contributions of our work is an specific solution which has a temporal complexity $O(n)$ using an iterative and greedy algorithm.

The **iterative step** main operation is the search of the most probable state of a new line, using the information provided by the previous lines. Be $Q_c(t)$ the set of known states in instant $t$, that is, all the previously visited states. Ideally, it is possible to keep the whole history for $Q_c(t)$, but in practice memory available is limited. Be $A_{t+1}$ the event *get a value for $Y(t + 1)$ as or more extreme than $y(t + 1)$*. We search for the state that maximizes the probability, $\arg\max_{q_k \in Q_c(t)} P(S(t + 1) = q_k \mid A_{t+1})$. Using Bayes Theorem we get that

$$P(S(t + 1) = q_k \mid A_{t+1}) \propto P(S(t + 1) = q_k) \cdot P(A_{t+1} | S(t + 1) = q_k) \tag{1}$$

where symbol $\propto$ allows the elimination of all constants not depending on $k$. As long as we don't know in advance the structure of states, it seems reasonable to suppose that all are equally likely and $P(S(t + 1) = q_k) = \frac{1}{N}$. But, on the other side, we can estimate $P(A_{t+1} | S(t+1) = q_k)$ because we are assuming a normal distribution for memory lines. Finally, we can simplify the expression:

$$P\big(S(t + 1) = q_k \mid A_{t+1}\big) \propto \Phi\left(-\frac{|y(t + 1) - \mu_k(t + 1)|}{\sigma_k(t + 1)}\right) \tag{2}$$

So

$$q_{mp}(t + 1) = \arg\max_{q_k \in Q_c(t)} P\big(S(t + 1) = q_k \mid A_{t+1}\big) = \arg\max_{q_k \in Q_c(t)} \Phi\left(-\frac{|y(t + 1) - \mu_k(t + 1)|}{\sigma_k(t + 1)}\right) \tag{3}$$

is the most probable state among visited, where $\mu_k(t + 1)$ and $\sigma_k(t + 1)$ are the mean and standard deviation for group $k$ in time $t + 1$ and whose estimation is described later in this section.

We still have to consider the possibility that the line comes from a new state not seen before. For this we stablish a maximum threshold ($p_{min}$) for the probability of belonging to $q_{mp}(t + 1)$. If $p_{max}(t + 1) = P(S(t + 1) = q_{mp}(t + 1) \mid A_{t+1})$, we assign the state using these rules

$$\widehat{S(t + 1)} = \begin{cases} q_{mp}(t + 1) & \text{if } p_{max}(t + 1) \geq p_{min} \\ \text{new state } q_n, \text{with } q_n \notin Q_c(t) & \text{if } p_{max}(t + 1) < p_{min} \end{cases} \quad (4)$$

By experimentation we checked that $p_{min} = 0.000025$ is reasonably good for most cases.

In order to find $q_{mp}(t + 1)$ according to expression 3, it is necessary to estimate $\mu_k(t + 1)$ and $\sigma_k(t + 1)$, $\forall q_k \in Q_c(t)$. Those estimations depend upon $n_k(t)$, which is the number of memory lines assigned to $q_k$ in $t$. Be $y_{k,1}, \ldots, y_{k,nk}(t)$ the memory lines assigned to $q_k$ in $t$. The estimation for $\mu_k(t + 1)$ and $\sigma_k(t + 1)$ is updated each time a new address is assigned to $q_k$ or when a new group is created. This procedure depends upon $n_k(t)$:

- If a new group is created, $n_k(t) = 1$, then the nearest group in terms of the mean is selected, that is, $q_{k'} = \arg\min_{q_i \in Q_c(t)} |y_{k,1} - \mu_i(t)|$. For a correct estimation of $\sigma_k$, as long as there is only one observation, its behavior is assumed to be similar than the behavior of the nearest group, but probably with larger variance. The reason is that the larger the variance is, the higher is the probability that the group grows, and that is why we multiply it by a factor of 10. The line is used for $\mu_k$ estimation.

$$\widehat{\mu_k(t')} = y_{k,1}, \quad \widehat{\sigma_k(t')} = \min(10\sigma_{k'}(t), \sigma_{max}), \quad t' \geq t \quad (5)$$

we use $\sigma_{max} = 200$ to prevent the value from growing excessively. In order to avoid that the new distribution overlap with other near distributions, it is recommended that $\widehat{\sigma_k(t)} \leq \frac{|\widehat{\mu_k(t)} - \widehat{\mu_{k'}(t)}|}{2}$. Otherwise, $\widehat{\sigma_k}$ is divided by 2 until the previous condition accomplishes.

- $1 < n_k(t) \leq 5$ means that the state has been previously created but currently their size is small. Because of this, estimation for $\sigma_k(t)$ doesn't change. $\mu_k(t')$ is estimated as described,

$$\widehat{\mu_k(t')} = \frac{1}{n_k(t)} \sum_{i=1}^{n_k(t)} y_{k,i}, \quad t' \geq t \quad (6)$$

- If $n_k(t) > 5$ then it is presumed that the size is enough for a linear regression model.

$$Y_{k,i} = \beta_0 + \beta_1 \cdot i + \epsilon_{k,i} \qquad i = 1, \ldots, n_k(t) \quad (7)$$

assuming $\epsilon_{k,i} \sim \mathcal{N}(0, \sigma_k^2)$. The Least Squares Method used here is widely known. This method applied to our particular problem directly allows for the following parameters estimation

$$\widehat{\beta_0} = \frac{1}{n_k(t) \cdot (n_k(t) - 1)} \sum_{i=1}^{n_k(t)} (2(n_k(t) + 1) - 6i) y_{k,i}$$

$$\widehat{\beta_1} = \frac{1}{n_k(t) \cdot (n_k(t) - 1)} \sum_{i=1}^{n_k(t)} \left(-6 + \frac{12i}{n_k(t) + 1}\right) y_{k,i} \quad (8)$$

Following again the Least Squares Method, $\sigma_k$ estimation satisfy that:

$$\widehat{\sigma}_k = \sqrt{\frac{1}{n_k(t) - 2} \sum_{i=1}^{n_k(t)} (Y_{k,i} - \widehat{\mu_{k,i}})^2} \qquad \widehat{\mu_{k,i}} = \widehat{\beta}_0 + \widehat{\beta}_1 \cdot i, \quad i = 1, \ldots, n_k(t) \qquad (9)$$

Finally,

$$\widehat{\mu_k(t')} = \widehat{\beta}_0 + \widehat{\beta}_1 \cdot (n_k(t) + 1) \qquad \widehat{\sigma_k(t')} = \begin{cases} 1 & \text{si } \widehat{\sigma}_k \leq 1 \\ \widehat{\sigma}_k & \text{si } 1 < \widehat{\sigma}_k < \sigma_{max} \\ \sigma_{max} & \text{si } \widehat{\sigma}_k \geq \sigma_{max} \end{cases} \qquad t' > t \qquad (10)$$

In practice it is better to just use the last observations because the adaptation to state changes is quicker and, also, less memory is needed.

The **first step** of this method is to identify the initial group. It is extremely important for the initial group identification to be correct, because all the described procedure is based on it. Consequently, as criteria must be restricted, all the observations are comparable. There are several possible heuristics that can be applied:

- Waiting in order to get several requests to consecutive lines. This is an strict requirement.

- Non-supervised clustering techniques. This is a softer requirement based on considering that a set of accesses belongs to the same group as long as the resulting dendogram built with a hierarchical method (like Ward Procedure [33]) does not show significant evidences of more than one group. For example, if $h_1, \ldots, h_n$ are the sorted dendogram heights, then the requirement could be $\frac{h_i}{h_{i-1}} < u_{ini}, i = 2, \ldots, n$ with $u_{ini}$ based on experimentation.

Once the first sequence of length $n$ starting at $t = i, Y(i), \ldots, Y(i + n)$, has been chosen, then we discard the old lines and calculate $\mu$ and $\sigma$:

$$\widehat{S(j)} = \emptyset, j = 1, \ldots, i - 1$$

$$\widehat{S(j)} = q_1, j = i, \ldots, i + n$$

$$\widehat{\mu_1(t')} = \frac{1}{n+1} \sum_{j=0}^{n} Y(i + j), \quad t' \geq i \qquad (11)$$

$$\widehat{\sigma_1(t')} = \sqrt{\frac{1}{n} \sum_{j=0}^{n} (Y(i + j) - \mu_1(i + j))^2}, \quad t' \geq i$$

Using the group classification provided by the Hidden Markov Model and the statistic models developed for each group, it is possible to easily extend this methodology into a prefetch technique. Suppose that in time $t$ a request to line $Y(t)$ is performed and assigned to group $\widehat{S(t)}$. The group defines the trend and behavior of its lines, so if there are enough observations to account the group as stable (usually $n_k(t) = 20, 50$ or $100$) then prefetches can be triggered.

We will presume that in the model defined in 7 the parameters have been previously estimated. Then, the prefetch predicted for the next observation assigned to the same group is $\mu_{k,n_k(t)+1} = \widehat{\beta}_0 + \widehat{\beta}_1 (n_k(t) + 1)$. This prediction represents a single line so the probability of

success is small. Instead, the prefetch mechanism proposed is based on a prediction interval with confidence $(1 - \alpha)$. This prediction interval when assuming a normal distribution with unknown variance uses the t-Student probability distribution, and applied to our particular case results in,

$$\widehat{Y_{k,n_k(t)+1}} \pm t_{n_k(t)-2,1-\frac{\alpha}{2}}\sqrt{\widehat{\sigma_k}^2\left(1 + \frac{2(2n_k(t)+1)}{n_k(t)(n_k(t)-1)}\right)} \tag{12}$$

The probability of success when all the lines in the interval 12 are prefetched is $1 - \alpha$. Prefetch mechanism triggers one request for each prefetched line to the NVM-RAM to store in the DRAM cache.

## Prefetcher implementation

The diagram in Fig 10 shows an schematic on-chip implementation of HMM described in section. On each LLC miss, the virtual address (VA) along with the PID (ASID) is sent to this circuit. The heuristic of this circuit is that the address may be linked to a group which is identified by one specific entry in the Group Table (GT) or, otherwise, a new group is created. If the address can be linked to a group, then the prefetch interval can be accurately generated. Two tables are needed with the same number of entries representing the maximum number of groups that can be simultaneously detected:

- Group Table (GT). Stores the parameters $\mu_i$ and $\sigma_i$ along with the LRU bits. Parameters $\mu_i$ and $\sigma_i$ characterize each group and allow, along with the last accesses on the group, to accurately generate the prefetch interval.

- History Table (HT). Stores the last memory accesses on each group, along with two fields used to maintain it as a circular list.

Let $q$ be the memory adress on a LLC miss. The circuit in Fig 10 performs two steps:

1. Group identification and group creation. LLC miss is compared with each entry in GT, calculating the entry (i) with $q_{mp} = \arg\max_i \Phi\left(\frac{|q-\mu_i|}{\sigma_i}\right)$ and the entry nearest to the current access (j) $(min|\mu_j - q|)$. If $q_{mp} \geq Prob_{min}$ then the current LLC miss is succesfully linked to a group (i) and the prefetch interval can be calculated. Otherwise, a new group is created based on the closest group identified by j.

2. Prefetch interval calculation. Once the group is identified, q is inserted in HT and then the entry (i) is read into a buffer which is used to calculate a new pair $\mu_i$ and $\sigma_i$ which is then stored in the same entry in GT. Using those new values, $\mu_i$ and $\sigma_i$, the prefetch interval is generated.

Complexity of the Prefetcher in Fig 10 depends on the maximum number of identifiable groups ($ng$) and the number of requests stored on each HT entry ($nh$). If tags are 6-Bytes (VA +ASID) long, then memory overhead is $6 \cdot ng \cdot nh + 8 \cdot ng + 6 \cdot nh - Bytes$. For typical values of $ng = 100$ (10-core processor with 10 groups identifiable on each thread) and $nh = 20$, memory overhead is 12.6-KBytes. Computation complexity is $O(ng + nh)$, therefore it is $O(n)$ complexity. Implementation price of our HMM proposal has two main sources, the use of VA addresses in the on-chip hierarchy and the implementation of the prefetch circuit. The circuit implementation in the example requires 12.6-KBytes of memory to which should be added the logical circuits to read/write the tables and perform simple calculations. Transistor count in today

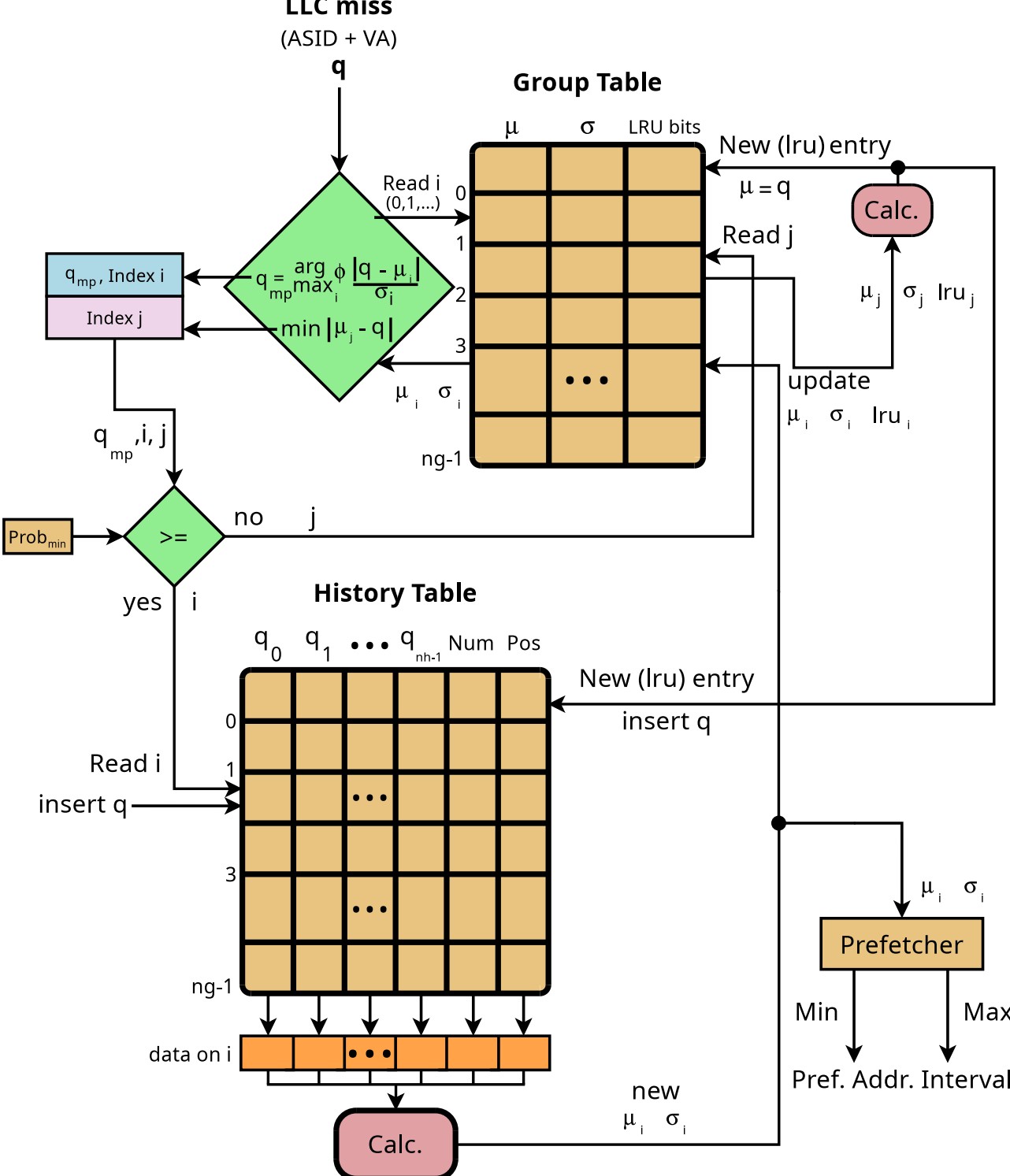

**Fig 10. Prefetch circuit.** Schematic description of the prefetcher on-chip implementation. LLC miss address (q) is used to identify a group and generate the prefetch address interval, or to create a new group based on the nearest current group. This implementation of HMM allows for precise identification of the simultanenous off-chip memory groups accessed by the different processes running in a multicore chip.

chips are over 10-billion so the circuit implementation cost can be considered negligible. The use of VA addresses increases 15% approximately the size of the tag area (about 1.4% increase in total cache size) with the corresponding increase in energy consumption. This is the main modification and the main source of price increase. Because this circuit is activated with low frequency (only LLC misses) and is not in the critical path, we consider that its implementation is feasible and with great potential advantages.

## Evaluation

In this section we describe the evaluation environment, the simulation configuration, benchmark description and the results obtained in our proposed prefetcher compared with similar hardware prefetchers in single-core, multicore (9 and 16 cores) and multiprogrammed multicore (4 cores with 4 benchmarks each) architectures.

### Evaluation setup

We evaluate our proposed HMM prefetch technique using Pin Tool [34] with an in-house trace-driven model of the memory hierarchy. The memory and prefetchers evaluation parameters are shown in Table 1 and a basic description of the 20 benchmarks used for evaluation appears in Table 2.

The evaluation of HMM prefetch has been done in two steps, on-chip and off-chip evaluation. In the first step the on-chip cache has been evaluated to get the off-chip memory access pattern of the benchmarks. We have evaluated a memory hierarchy with four on-chip LLC configurations as appears in Table 1-Memory. The results of this evaluation have been presented in the section *Off-chip Memory Accesses*. We show in this section the main characteristics of spatial, temporal and algorithmic off-chip locality and perform a detailed analysis. Our main conclusion is that off-chip access patterns are very different and much more complex than on-chip accesses so specific hardware prefetch solutions can be applied.

In the second step we use the best LLC configuration (size 32MB, associativity 8) to evaluate HMM and to compare it with similar proposals, analyzing its impact on a DRAM cache over a backup NVM-RAM memory. The size and associativity of DRAM cache has been chosen to prevent the working set of the applications from fitting. The reason to do this is that it is difficult to perform an evaluation with realistic gigabit-size workloads, so we have opted for reducing the size of external DRAM cache to reproduce the behavior of a bigger working set size with a bigger cache memory size. See memory simulation configuration on Table 1-Memory.

**Table 1. Evaluation parameters.**

| | |
|---|---|
| **Memory** | |
| L1 Cache | Instruction: size 32KB, Line 32B, Assoc. 32, private |
| | Data: size 32KB, Line 32B, Assoc. 32, private |
| L2 Cache | Size 2MB, Line 64B, Assoc. 1, private |
| LLC Cache | Size 16MB/32MB, Line 64B, Assoc. 1/8, shared |
| Off-chip Cache | Size 16MB, Line 64B, Assoc. 4, shared |
| **Prefetchers** | |
| HMM | History size (nh) 20, number of groups (ng) 180, size 22,6KB |
| G/DC | GHB size 512, history length 4, prefetch degree 4, size 8KB |
| G/AC | GHB size 512, history length 4, prefetch degree 4, size 8KB |

Memory hierarchy geometry used in the evaluation and parametrization of prefetchers.

**Table 2. Benchmarks.**

| Integer | | Floating Point | |
|---|---|---|---|
| bzip2 | Compression | milc | Quantum Chromodyn |
| gcc | C Compiler | namd | Molecular Dynamics |
| mcf | Combinatorial Optimiz. | povray | Ray Tracing |
| hmmer | Search Gene Sequenc. | lbm | Fluid Dynamics |
| sjeng | Chess | sphinx3 | Speech Recognition |
| libquantum | Quantum Computing | Other | |
| h264ref | Video Compression | Firefox | Web Browser |
| omnetpp | Discrete Event | g++ | C++ Compiler |
| astar | Path Finding | Openoffice | Text Processing |
| xalan | XML Processing | | |
| gobmk | Artificial Inteligence | | |
| specrand | Random Generator | | |

Seventeen out of the twenty applications we use for evaluation came from SPEC CPU2006 [35] and the remaining three are specially selected applications to get different memory patterns (see Table 2). We used the full SPEC CPU2006 benchmark suite, except for those applications coded in Fortran and *perlbench* where we encountered compilation issues. *Specific* applications are a web browser, a text processing tool and a C++ compiler. What we look for are both regular and irregular memory patterns. For each execution we take 20-50M memory access requests on the steady phase. Software prefetches from the compiler are included in the working set as regular LLC misses. We decided to keep software prefetches because our hardware prefetcher could take advantage of the hints from the more complex heuristics implemented by software.

We compare our proposal with two prefetchers which have very good performance in conventional memory hierarchies: the Global Delta-Correlation (G/DC) and the Global Address-Correlation (G/AC), flavors of the Global History Buffer (GHB) proposal [22]. G/AC is an implementation of the Markov Predictors [20] based on miss address sequences, while G/DC is a distance prefetching method based on difference between miss addresses. Table 1 shows the values used for the different sizes of the data structures needed for HMM proposal, and, also, the size of the tables of the two high performance prefetchers used to compare. For HMM the number of groups (NG) is 180, which apply for a 9-core processor detecting 20 groups on average per application. The size of the chain used for group identification (NH) is 20. For G/DC and G/AC we use a Global History Buffer (GHB) of 512 entries and an Index Table (IT) of the same size.

Latency and bandwidth in backup memories (NVM-RAM) are very dependent on hit ratio in the corresponding caches. That is why in this section we evaluate hit ratio of off-chip DRAM cache with our HMM prefetch technique and compare it against BASE (no prefetch), G/DC and G/AC proposals. We also evaluate the *prefetch accuracy* defined as the fraction between the number of useful prefetch requests (requests providing data that is used in the near future) and the number of total prefetch requests. In the figures we represent *accuracy* using *overhead*, that is, the percentage of unnecessary requests triggered by the prefetcher, which involves a waste of bandwidth and energy.

## Single-core results

In this section we evaluate hit ratio and accuracy for each one of the applications running in a one-core processor. A simulation is performed for each application to check the adaptability of

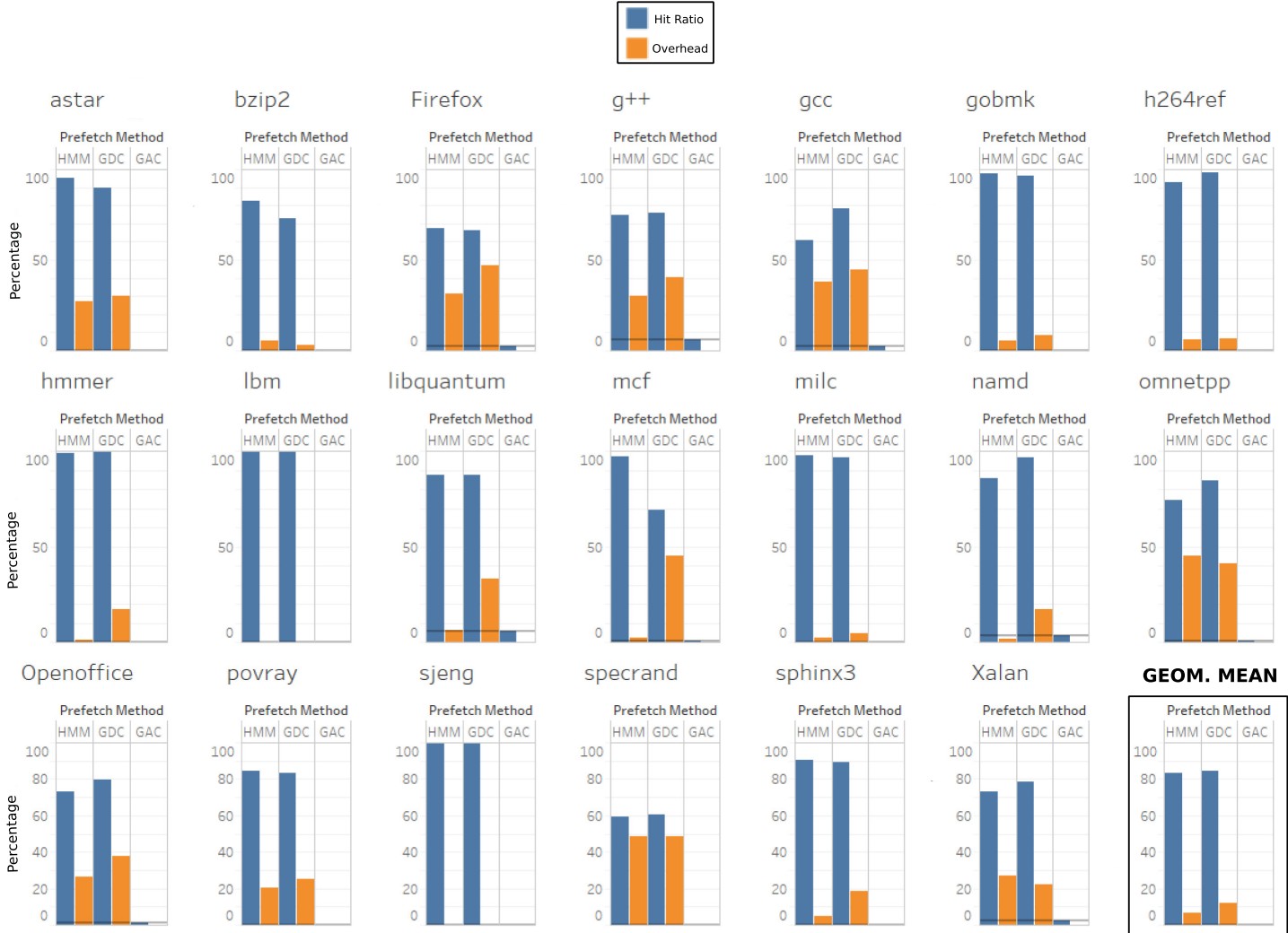

**Fig 11. Hit ratio and overhead of BASE, HMM, G/DC and G/AC in a single core architecture.** The hit ratio for the BASE experiment is represented by a horizontal line (0-6.3% for all benchmarks). The last plot is the geometric mean of all benchmarks.

our proposal to the data complexity of each application. Fig 11 shows the results for the 20 applications with single-core evaluation. The last plot represents the geometrical mean of the 20 applications. BASE hit ratio is represented with a horizontal line that cannot be seen on most of the applications because it is very low (0-6.3%), i.e. the cache without any prefetching is very inefficient which confirms our study in section *Off-chip Memory Accesses*. This is explained by the fact that most of the spatial and temporal locality has been filtered in on-chip caches. The same explanation can be applied to G/AC proposal in which the use of memory addresses cannot capture the complexity of multiple simultaneous data patterns, and hence its hit ratio is near zero for most of the applications.

Results with HMM and G/DC show a significant increase of hit ratio, to the range 60-99%. This shows the success of these prefetchers to detect most of the memory accesses, and thus the relevance of prefetch techniques in off-chip memory devices. Applications with regular data access patterns have good hit ratio with any of the two prefetch techniques, HMM and G/DC. Applications with multiple simultaneous memory areas and complex access patterns, like

*astar*, *bzip2* and *mcf*, show a better hit ratio with our proposal HMM. The biggest difference appears in *mcf* which is an implementation of *Network Simplex Algorithm*. This implementation requires a vector of simplex multipliers to be maintained on each feasible solution, that is, on each step. This increases the amount and complexity of data structures, and also the complexity of data access patterns. HMM adapts well to this complexity and to the multiple simultaneous memory areas involved at the same time on each iteration. *astar* implements path-finding algorithms which travel along graphs that map regions with neighborhood relationships, so data access pattern is complex and HMM adapts better than its competitors to it. On average, the efficiency of HMM and G/DC is very similar, being G/DC slightly superior.

Fig 11 also shows the prefetch *overhead* for each application. Overall, HMM has better results for most of the benchmarks. This can be observed specially in *mcf*, *hmmer*, *libquantum* and *firefox*. Average *overhead* of G/DC is 12.5% while HMM *overhead* is 6.5%, which represents a 48% improvement. The single-core results show that HMM adapts very well to complex applications. DRAM cache hit ratio is very similar to G/DC but *overhead* is significantly reduced. It is important to check if this trend is maintained when complexity increases.

## Multicore results

Real off-chip memory systems must handle the joint complexity of all the cores in the chip, so in this section we evaluate a multicore architecture to get complete data of how our HMM proposal deals with the increased data access patterns complexity. In the first set of multicore experiments we use a 9-core architecture running one application each. The evaluated prefetchers receive a set of LLC misses from the simultaneous running of 9 different benchmarks, each of which has a complex spatial, temporal and algorithmic complexity.

Fig 12(a) represents hit ratio and *overhead* for our multicore processor running four different *application mixes*. The *application mixes* have been chosen to create a balanced merge of complex and simple data pattern applications. As shown in the figure, HMM hit ratio is better

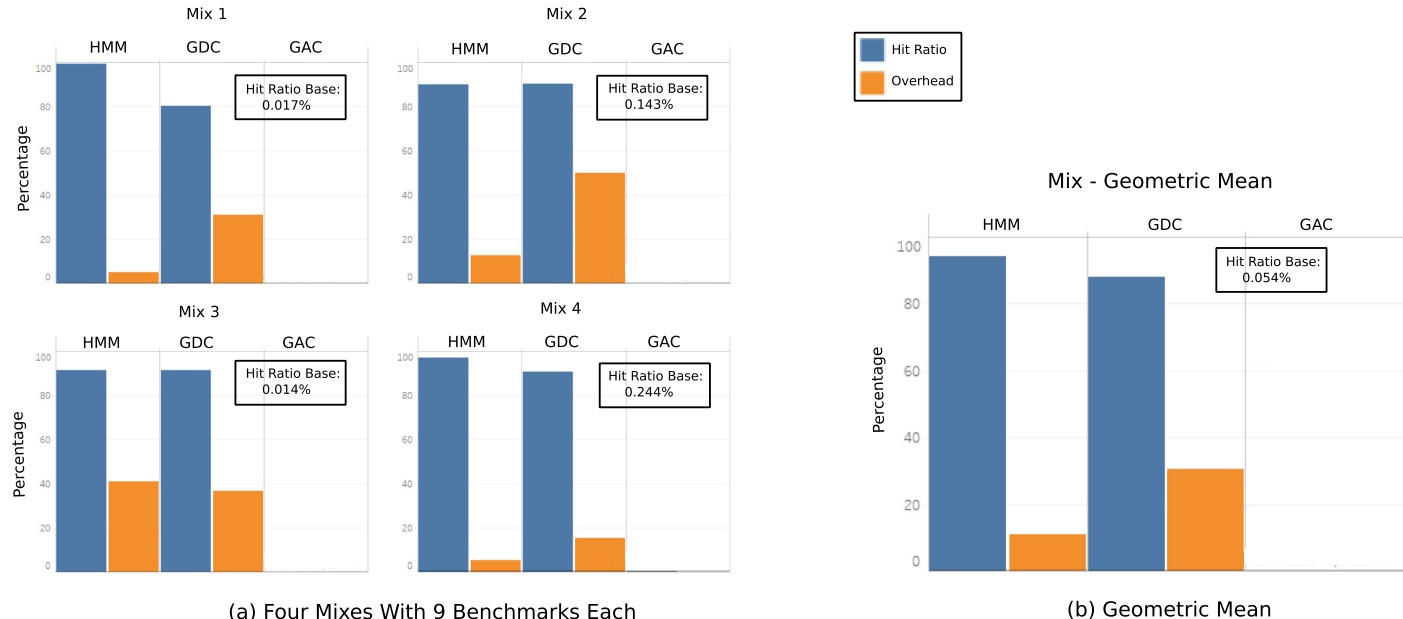

(a) Four Mixes With 9 Benchmarks Each    (b) Geometric Mean

**Fig 12. Hit ratio and overhead of BASE, HMM, G/DC and G/AC in a 9 core architecture.** The hit ratio of the base experiment is shown in the box. Each mix consists of nine benchmarks.

than G/DC for all the *application mixes*. On average, see Fig 12(b), HMM hit ratio is 94% versus 88% of G/DC (7.2% improvement). Accuracy is much better in HMM, with a 10.8% average *overhead*, versus 30.5% of G/DC, leaving HMM with a 64% improvement over G/DC. As well as in single core simulations, BASE and G/AC behave poorly in multicore simulations with a hit ratio of $\simeq 0.05\%$.

For the second set of experiments the goal is to check the behavior of HMM and G/DC when the complexity of the cache miss address patterns is greatly enlarged. For this reason we run 16 benchmarks on a 16-core system and we keep the table size for HMM and G/DC and, also, the DRAM cache size and associativity. This way we can simulate a drastic increase in memory pressure in the off-chip memory system. The results for a new mix of 16 benchmarks are shown in Fig 13(a). As in the 9-core experiment, hit ratio for the BASE experiment is very low ($\simeq 0.04\%$), it is much better for G/DC (55%) and it is very good for HMM (97%). This means that, while both G/DC and HMM are successfull prefetch proposals for an off-chip cache, HMM is the one that best identifies the accesed memory lines, with a 76% improvement over G/DC. *Overhead* is 9% for HMM and 54% for G/DC (83% improvement) which means that our proposal is more precise than G/DC identifying the multiple groups of addresses so the generated prefetches are useful with much greater frequency in HMM than in G/DC. Multicore experiments show how HMM adapts well when data patterns complexity increases, getting better hit ratio and *overhead* than similar proposals.

## Multiprogrammed multicore results

The next set of experiments aims to check how our proposal deals with a more realistic and complex run environment with more processes than cores in which the operating system introduces changes in address patterns. We have designed an experiment with a multicore (four core) architecture in which each of the cores executes four benchmarks using a round robin multiprogramming algorithm. OS generated addresses are not included in the experiments. The memory access patterns will abruptly and radically change when a trap occurs because the core switches to a different process. Thanks to this experiment, it can be analyzed the complexity induced by the operating system when several benchmarks share the same core. Results of this experiment are shown in Fig 13(b). Hit ratio is $\simeq 0.3\%$ in the BASE experiment, 87% in G/DC, and 92% in HMM while overhead is 27% in G/DC and 13% in HMM. These results are similar to those obtained in the multicore experiment, confirming the conclusions

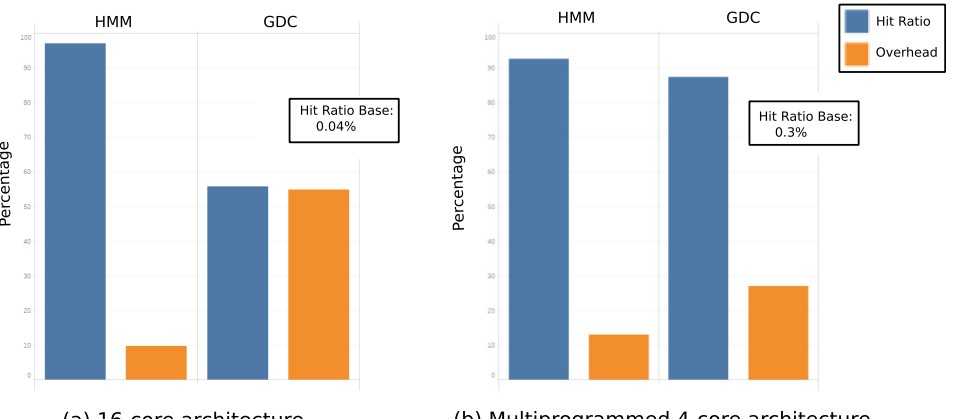

(a) 16-core architecture (b) Multiprogrammed 4-core architecture

**Fig 13. Hit ratio and overhead of BASE, HMM and G/DC in a 16 core architecture and in a multiprogrammed 4 core architecture.** The hit ratio of the base experiment is shown in the box.

previously expressed in which our proposal identifies a greater number of effectively accesed lines and a smaller number of unused lines than G/DC.

The main conclusion that can be drawn from the evaluation of our technique is that it adapts very well to the complexity of off-chip accesses, converting an ineffective cache into a cache with very good hit ratio. HMM also gets better results than similar proposals, improving hit ratio up to 76% and overhead up to 83% over G/DC.

## Conclusions

Current on-chip cache hierarchies are very efficient capturing spatial and temporal locality but, as an unintended consequence, off-chip accesses become very complex and difficult to predict. This is an important problem when using a DRAM cache to hide the latency of a high capacity, high density and low energy NVM-RAM, because the efficiency of the external DRAM cache degrades greatly.

This paper presents a specific prefetch mechanism for off-chip memory accesses. The first contribution of this study is the characterization of external memory access patterns to extract the main properties that guide the development of our contribution, which is a novel prefetch technique. This prefetch technique, based on a Hidden Markov Model, allows for the simultaneous identificacion and tracing of multiple groups of access patterns directly related to algorithmic locality. HMM enables the management of very complex off-chip memory access patterns, triggering the prefetch operations on each group independently so it improves the hit ratio of external DRAM cache. We have modelled and evaluated our prefetch proposal using traces from real benchmarks executions.

The results show that HMM behaves better when complexity increases, thus, running better in multicore than in single core simulations. HMM hit ratio improves up to 76% in multicore simulations when compared with conventional prefetch techniques. Accuracy of prefetches is very important because it reduces undesired and useless off-chip data traffic (*overhead*). HMM reduces *overhead* over existing current prefetchers by a 48% in single core simulations and up to 83% in multicore simulations. Therefore our proposal improves external DRAM cache hit ratio while reducing mistaken prefetch requests, optimizing the critical off-chip memory traffic.

## Author Contributions

**Conceptualization:** Benjamín Sahelices.

**Data curation:** Adrián Lamela, Benjamín Sahelices.

**Formal analysis:** Adrián Lamela, Benjamín Sahelices.

**Funding acquisition:** Benjamín Sahelices.

**Investigation:** Adrián Lamela, Benjamín Sahelices.

**Methodology:** Adrián Lamela, Óscar G. Ossorio, Guillermo Vinuesa, Benjamín Sahelices.

**Project administration:** Benjamín Sahelices.

**Resources:** Benjamín Sahelices.

**Software:** Adrián Lamela, Benjamín Sahelices.

**Supervision:** Benjamín Sahelices.

**Validation:** Adrián Lamela, Óscar G. Ossorio, Guillermo Vinuesa, Benjamín Sahelices.

**Visualization:** Adrián Lamela, Óscar G. Ossorio, Guillermo Vinuesa.

**Writing – original draft:** Óscar G. Ossorio, Guillermo Vinuesa, Benjamín Sahelices.

**Writing – review & editing:** Óscar G. Ossorio, Guillermo Vinuesa, Benjamín Sahelices.

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
