## [Decision Letter · Decision Letter 0]

9 Jun 2021

PONE-D-20-36013

Off-chip Prefetching Based on Hidden Markov Model for Non-Volatile Memory Architectures

PLOS ONE

Dear Dr. Sahelices,

Thank you for submitting your manuscript to PLOS ONE. After careful consideration, we feel that it has merit but does not fully meet PLOS ONE’s publication criteria as it currently stands. Therefore, we invite you to submit a revised version of the manuscript that addresses the points raised during the review process.

We look forward to receiving your revised manuscript.

Kind regards,

Anandakumar Haldorai, PhD

Academic Editor

PLOS ONE

Additional Editor Comments (if provided):

The technical concept presented in the research paper is an important one, in order to improve the Hidden Markov Model for Non-Volatile Memory Architectures.

- As a special emphasis is made on the adoption of Markov-based technique. The paper would benefit with an extended discussion on appropriate models for Non-Volatile Memory, what has been considered in the literature and why Markov-based technique is the best option?

- Is it reasonable to use LLC for evaluating the performance in HMM prefetch? Which scenarios are the authors considering? Also, for the case of nvm-ram memory, which are the corresponding scenarios? A discussion on this issue should be added.

Journal Requirements:

Reviewers' comments:

Reviewer's Responses to Questions

**Comments to the Author**

1. Is the manuscript technically sound, and do the data support the conclusions?

Reviewer #1: Yes

Reviewer #2: No

Reviewer #3: Yes

2. Has the statistical analysis been performed appropriately and rigorously? 

Reviewer #1: Yes

Reviewer #2: No

Reviewer #3: Yes

3. Have the authors made all data underlying the findings in their manuscript fully available?

Reviewer #1: Yes

Reviewer #2: No

Reviewer #3: Yes

4. Is the manuscript presented in an intelligible fashion and written in standard English?

Reviewer #1: Yes

Reviewer #2: No

Reviewer #3: Yes

5. Review Comments to the Author

Reviewer #1: 1. The manuscript is written in well structured form.

2. Latest approaches can be included in the related work (2020 & 2021)

3. Provide proof for the performance of HMM when complexity increased

4. More explanation required for HMM in prefetching

Reviewer #2: This study presents an off-chip prefetch technique that deals specifically with the algorithmic complexity of off-chip memory access patterns, based on the design, modeling and evaluation of a prefetch technique based on a hidden Markov model.

In general, the writing of this work is confusing and redundant, more than 50% of the literature presented is old and generally of low quality, the figures presented are not very legible, lack homogeneity and are poorly described in the text. There is a lot of information that is not justified with bibliographic information in the text, such is the case of the equations presented where at no time is the reference source specified. The idea is not bad, but it is necessary to restructure the document. For all of these reasons, I would then not recommend this work for publication.

Major comments:

The abstract is very extensive and quite room for improvement.

The use of the language is misused.

It is necessary to do a discussion section of the results.

The results framework must be considerably improved since the structure it presents does not allow the contribution to be clearly visualized.

P1L1: The introduction section should have more current information, it would be convenient to restructure it to highlight the novelty of the work. What is the real problem, as a result of the bibliographic research?

P5L153: It is necessary to make a contrast with respect to current works, since in this way the novelty of the work is questioned.

P7L218: It is convenient to improve the description of mathematical expressions, in the same way whenever a mathematical expression is placed it is necessary to specify the reference.

Minor comments:

Figures and tables do not have an adequate format, they are not correctly described and there is a lack of homogeneity between them.

It would be interesting to contrast the results obtained with respect to another modeling technique and contrast them with respect to some performance index.

P5L153: The “Related work” section should be described in the introduction.

Reviewer #3: The technique you presented in the research paper is an important one, in order to improve the non-volatile memory technology. I appreciate your effort. But I wonder if that could rise the price and how much. Nowadays this is an important criteria also. You computed also the impact of your improvement in the technical parameters but in my opinion is not enough in this field. Maybe you could add also some phrases on this too in your manuscript. This could improve the impact of your research especially in the production field because our research is also dedicated to be applied.

6. PLOS authors have the option to publish the peer review history of their article (what does this mean?). If published, this will include your full peer review and any attached files.

Reviewer #1: **Yes: **Ascar Davix X

Reviewer #2: No

Reviewer #3: No

---

## [Author Response · Author response to Decision Letter 0]

1 Aug 2021

All the answers to the reviewers and editor comments have been included in the file "Response_To_Reviewers.pdf" and they appear included in the differences document and in the final version of our manuscript.

---

## [Decision Letter · Decision Letter 1]

23 Aug 2021

Off-chip Prefetching Based on Hidden Markov Model for Non-Volatile Memory Architectures

PONE-D-20-36013R1

Dear Dr. Sahelices,

We’re pleased to inform you that your manuscript has been judged scientifically suitable for publication and will be formally accepted for publication once it meets all outstanding technical requirements.

Kind regards,

Anandakumar Haldorai, PhD

Academic Editor

PLOS ONE

Additional Editor Comments (optional):

The changes have been made accordingly, Recommended for further publication process.

Reviewers' comments:

Reviewer's Responses to Questions

**Comments to the Author**

1. If the authors have adequately addressed your comments raised in a previous round of review and you feel that this manuscript is now acceptable for publication, you may indicate that here to bypass the “Comments to the Author” section, enter your conflict of interest statement in the “Confidential to Editor” section, and submit your "Accept" recommendation.

Reviewer #1: All comments have been addressed

Reviewer #3: All comments have been addressed

2. Is the manuscript technically sound, and do the data support the conclusions?

Reviewer #1: Yes

Reviewer #3: Yes

3. Has the statistical analysis been performed appropriately and rigorously? 

Reviewer #1: Yes

Reviewer #3: Yes

4. Have the authors made all data underlying the findings in their manuscript fully available?

Reviewer #1: Yes

Reviewer #3: Yes

5. Is the manuscript presented in an intelligible fashion and written in standard English?

Reviewer #1: Yes

Reviewer #3: Yes

6. Review Comments to the Author

Reviewer #1: I appreciate the authors that all the suggestions are discussed neatly and modified the manuscript accordingly.

I recommend for publication of this manuscript.

Reviewer #3: I appreciate that you made corrections in the manuscript and you improved the content of it. The results of the research is better described now and more clearly for the readers. You improved the abstract by adding supplementary comments, probably suggested by the reviewers and this added value to the article. I remarked also that you expressed in a different manner the improvement that you obtained after the simulation. The references are now better used in the text. I express my appreciation for your effort to improve the manuscript and I agree with you modification that brings a more clear understanding of your research.

7. PLOS authors have the option to publish the peer review history of their article (what does this mean?). If published, this will include your full peer review and any attached files.

Reviewer #1: **Yes: **Dr.X. Ascar Davix

Reviewer #3: No

---

## [Editor Report · Acceptance letter]

3 Sep 2021

PONE-D-20-36013R1 

Off-chip Prefetching Based on Hidden Markov Model for Non-Volatile Memory Architectures 

Dear Dr. Sahelices:

I'm pleased to inform you that your manuscript has been deemed suitable for publication in PLOS ONE. Congratulations! Your manuscript is now with our production department. 

Kind regards, 

on behalf of

Dr. Anandakumar Haldorai 

Academic Editor

PLOS ONE